# Naked mole-rat mortality rates defy Gompertzian laws by not increasing with age

**J Graham Ruby, Megan Smith, Rochelle Buffenstein***

Calico Life Sciences LLC, South San Francisco, United States

**Abstract** The longest-lived rodent, the naked mole-rat (*Heterocephalus glaber)*, has a reported maximum lifespan of >30 years and exhibits delayed and/or attenuated age-associated physiological declines. We questioned whether these mouse-sized, eusocial rodents conform to Gompertzian mortality laws by experiencing an exponentially increasing risk of death as they get older. We compiled and analyzed a large compendium of historical naked mole-rat lifespan data with >3000 data points. Kaplan-Meier analyses revealed a substantial portion of the population to have survived at 30 years of age. Moreover, unlike all other mammals studied to date, and regardless of sex or breeding-status, the age-specific hazard of mortality did not increase with age, even at ages 25-fold past their time to reproductive maturity. This absence of hazard increase with age, in defiance of Gompertz's law, uniquely identifies the naked mole-rat as a non-aging mammal, confirming its status as an exceptional model for biogerontology.

DOI: https://doi.org/10.7554/eLife.31157.001

## Introduction

The longest-lived rodent, the naked mole-rat (Ctenohystrica, *Heterocephalus glaber*) is recognized as an animal model of biogerontological interest (*Buffenstein, 2005*; *Austad, 2010*; *Edrey et al., 2011*; *de Magalhães et al., 2007a*) with a maximal lifespan of more than 30 years in our captive care (*Lewis and Buffenstein, 2016*). This maximum lifespan is five-fold greater than predicted allometrically for a 40 g rodent (*Edrey et al., 2011*). Even in the wild, naked mole-rats are also considered to be very long-lived rodents with reports of breeding females remaining 17 years in the same colony while non-breeders remain 2–3 years (*Hochberg et al., 2016*). In sharp contrast, other rodents generally last only a season in the wild, with their young-age deaths attributed to predation, inclement weather, parasites, pathogens and lack of food resources (*Morgen, 2003*; *David and Jarvis, 1985*; *Boonstra and Krebs, 2012*; *Krebs et al., 1995*; *Krebs et al., 1973*). Laboratory mice, cosseted in a protected environment with *ad libitum* resources, live almost an order of magnitude longer than their wild counterparts. Nevertheless, these small mammals live only half as long as expected on the basis on body size and lifespan is limited by age-related, exponentially increasing intrinsic risk of dying (*Miller et al., 2002*). Similarly, the difference between captive and wild longevity records for naked mole-rats likely reflects a lower incidence of extrinsic rather than intrinsic mortality in captivity. What determines naked mole-rat lifespans in captivity is currently unknown.

The naked mole-rat is a strictly subterranean, mouse-sized rodent and is one of only two known eusocial mammals; it lives in large family groups (colonies) of up to 300 individuals (average colony size = 60). Their underground lifestyle in the arid and semi-arid regions of sub-Saharan Africa poses numerous ecological and physiological challenges, such as difficulty locating food underground, impaired heat and gas exchange, low oxygen availability, and exposure to heavy metals and other chemicals in soil and plant-defense toxins (*Buffenstein, 2000*). But this milieu also protects against

*For correspondence:
rbuffen@calicolabs.com

Competing interest: See
page 14

Reviewing editor: Michael
Rose, University of California,
Irvine, United States

dangers such as predation and above-ground climate extremes (e.g., cold, wind and rain), thereby providing several survival advantages (*Buffenstein, 2005*).

Breeding in each colony is monopolized by only one female (the 'queen') and 1 to 3 males at a time, with most animals in the colony showing no sexual behaviors throughout their life, but rather helping with the overall maintenance and survival of the colony, including care of pups (*Jarvis, 1981*). Queen status is primarily maintained through physical intimidation of subordinates and should the queen die or a non-reproductive subordinate be removed from the colony, subordinates can become reproductively active (*Clarke and Faulkes, 1997*). Attainment of breeder-status for either sex, regardless of age, is accompanied by morphological, physiological and behavioral changes, including changes in body size, brain volume and reproductive organ morphology (*O'Riain et al., 2000*; *Holmes et al., 2007*; *Peroulakis et al., 2002*). These differences are pronounced and obvious in the breeding female (*O'Riain et al., 2000*).

While the underlying determinants of species' longevity remain unknown, several physiological traits correlate with longevity (*Harvey and Purvis, 1999*). Across vertebrates, adult body mass is a strong correlate, with larger species living longer than smaller species. Naked mole-rats live exceptionally long vis-à-vis this metric: almost five-fold longer than expected (*Hulbert et al., 2007*). Species' ages of reproductive maturity ($T_{sex}$) are also predictive of their lifespans (*Hamilton, 1966*; *Rose et al., 2007*; *Prothero, 1993*). The definition of $T_{sex}$ is more nuanced than that of body weight: ambiguity exists around how the age of sexual maturity should be marked, whether $T_{sex}$ should include gestation time, and how $T_{sex}$ relates to other developmental milestones. $T_{sex}$ also lacks independence from both body weight and phylogeny. Nonetheless, it retains predictive power even when those variables and nuances are taken into account (*de Magalhães et al., 2007b*). Relative to $T_{sex}$, naked mole-rats are once again long-lived outliers: despite a long gestation length (66–76 days) and a six-month elapse from birth to reproductive capacity (*Edrey et al., 2011*), they live over three-fold longer than expected by this metric (*de Magalhães et al., 2007b*).

Beyond lifespan, the physiological declines that accompany advancing age in most mammals fail to manifest in naked mole-rats (*Buffenstein, 2008*). Breeding females show no menopause, retaining high fertility even at ages past 30 years (*Lewis and Buffenstein, 2016*). Neurogenesis is also prolonged and may continue for at least two decades (*Holmes, 2016*), and over a similar time course, no significant changes in cardiac function (*Grimes et al., 2014*), body composition, bone quality, and metabolism (*O'Connor et al., 2002*) are evident. Age-associated chronic diseases such as cancer are also rare (*Edrey et al., 2011*; *Lewis and Buffenstein, 2016*). Within the cell, proteasome function, as well as mitochondrial mass, gene and protein expression are maintained with age (*Pérez et al., 2009*; *Kim et al., 2011*). These molecular and physiological observations reflect broader phylogenic trends with respect to mortality and life history traits: species from safer habitats tend to live longer; those from more dangerous habitats tend to mature rapidly, reproduce early and die young (*Ricklefs, 1998*; *Pianka, 1970*).

The concepts of mortality and physiological decline associated with aging can be connected within the mathematical framework of the Gompertz-Makeham law of mortality (*Gavrilov and Gavrilova, 1991*). Age-specific hazard h(t) is defined by the equation $h(t) = \alpha e^{\beta t} + \gamma$. The first term ($\alpha e^{\beta t}$) describes the Gompertzian component of hazard, originally derived from examination of human life tables (*Gompertz, 1825*). This hazard increases exponentially with age, presumably due to intrinsic age-related physiological declines. In this framework, $\beta$ defines the rate-of-aging with respect to mortality: its value determines the doubling time for mortal hazard (for humans, ~7–8.5 years; [*Finch et al., 1990*]). The second term ($\gamma$) describes the Makeham component: a constant, extrinsically-imposed minimum hazard that can obfuscate Gompertzian hazard at younger ages (*Makeham, 1860*). The coefficient of the Gompertz term ($\alpha$) influences the age at which intrinsic hazard overtakes extrinsic hazard.

While naked mole-rats are already noted as exceptionally long-lived (*Buffenstein and Jarvis, 2002*), this status relies on small-sample-based estimates, leaving much uncertainty as to how exceptional their longevity may truly be and how they differ from other mammals with respect to the Gompertz-Makeham aging framework. Here, we compiled a large, historical dataset of naked mole-rat lifespans using records kept throughout the ~35 year maintenance of our collection (described in [*Edrey et al., 2011*]). With over 3000 data points, we constructed survival curves and performed analyses of age-specific hazard. In these analyses, this mouse-sized rodent exhibited no increase in

mortality hazard, that is, no Gompertzian aging, across its full, as-yet-observed, multi-decade lifespan. This life-history trend is unprecedented for mammals.

## Results

### Reports of naked mole-rat maximal lifespan have been based on observation time, not mortality

Our data set drew from 3848 historically catalogued naked mole-rats from our own collection. Of these, 3329 had high resolution (day/month/year) records of date-of-birth and date-of-death/censorship, allowing them to be included in our initial, most stringent analysis (*Figures 1* and *2*). Of the 519 excluded animals, 269 had lower-resolution data with respect to birth or death, and were optionally added to the analyses, but nevertheless yielded similar results (*Figure 3*). Right-censorship was applied in three cases: if the animal was (1) transferred to another collection, with the date of transfer considered as the censorship event; (2) euthanized for research purposes when still healthy, with the date of sacrifice as the censorship event; or (3) still alive when the process of compiling records was completed, with the date of completion as the censorship event (see Materials and methods). Our analyses of survival and mortality hazard began at $T_{sex}$ for naked mole-rats: 6 months (183 days; [*Edrey et al., 2011*]). Prior to $T_{sex}$, population data were incomplete, as juvenile animals only were given a unique ID (RFID) when more than 90 days old: the data for 30 animals that had died (n = 22) or been censored (n = 8) prior to 183 days were not considered here, leaving a population of 3299 animals for analysis.

Of the 3299 animals used to generate the survival plot and hazard estimates, only 447 had died (*Figure 1A*). The last recorded death was at 6529 days, at which point 23 animals remained alive; those were censored over the next ~15 years, with the last censorship occurring after a total of 11,077 days-of-life (30.3 years). The construction of a Kaplan-Meier survival curve for naked mole-rats starting at their $T_{sex}$ (183 days) revealed 61.6% of the population to have survived across the 30 year timespan covered by the data set (*Figure 1A*). This survival estimate suggested the naked mole-rat maximal lifespan to be far longer than the current 30 year published record (*Lewis and Buffenstein, 2016*; *Buffenstein, 2008*).

### The hazard of naked mole-rat mortality did not increase as a function of age

Using our survival data, we calculated age-specific mortality hazard across the scope of that data (*Figure 1B*). Estimate windows were expanded along the time axis, as the population diminished, so as to maintain approximately-even confidence intervals. Although slightly higher in the months immediately following $T_{sex}$ (d183-400), hazard remained consistently low thereafter, with the estimated probability of death per day never exceeding 1/10,000. Low hazard persisted for more than two decades, including in the final estimate bin, which was estimated using the final recorded death (including that event, at day 6529) and all subsequent censorship data, and for which mortality hazard was estimated to be $1.8 * 10^{-5}$ per day. This result was highly consistent with data we determined using life history information presented in (*Sherman and Jarvis, 2002*) for ≥15 year-old naked mole-rats in captivity: that paper reported the age at which twelve out of 86 animals died. Using the data reported there divided by the amount of observation time beyond 15 years-of-age produce an estimated probability of death per day of ~1/12,000; very similar to our estimate.

### Inclusion of low-resolution birth/death records reinforced constant-hazard demography

Our historical records included some data from 3848 naked mole-rats from our collection of which only 3329 high resolution data points were included in the most stringent analyses shown in *Figure 1*. The remaining 519 were excluded because data was either completely missing in our records (229 animals with missing birth, 20 listed as dead but with date-of-death listed as 'unknown'), improperly-formatted for interpretation (1 animal), or given at a less-than single-day resolution (269 animals). In the last case, it was possible to include those individuals in the analysis, albeit with the caveat of those new data being less precise. We repeated our analysis thrice below, including approximate data at three levels of stringency.

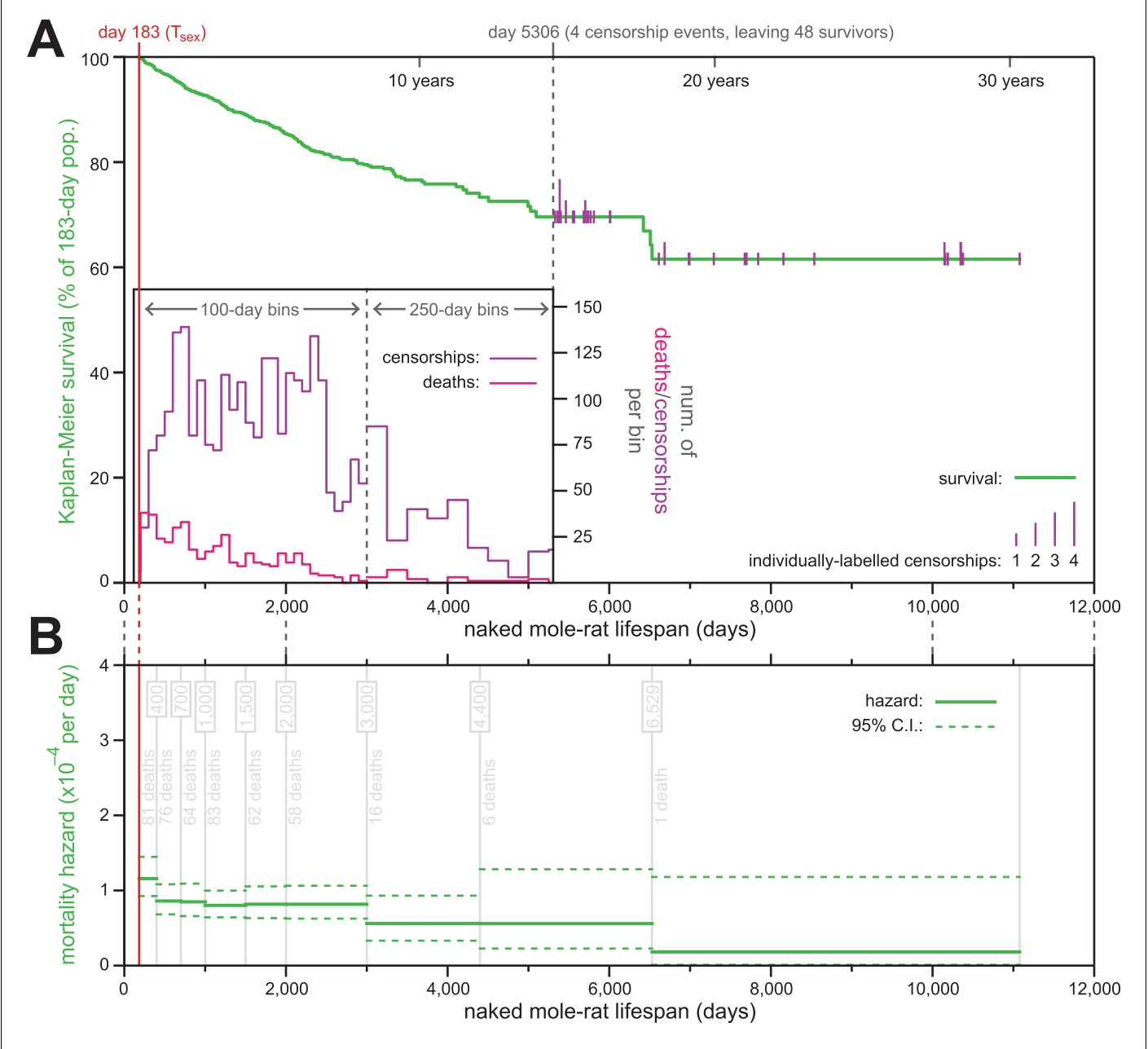

**Figure 1.** The mortality hazard of naked mole-rats failed to increase for at least 18 years. (**A**) Kaplan-Meier survival curve for naked mole-rats (green) after reaching reproductive maturity ($T_{sex}$; 6 months from birth; 183 days; red). The surviving population size drops below 50 at day 5306 (of 52 animals surviving to that day, four are censored: grey). After that, censorship events are indicated as vertical ticks (purple), the size of which is proportional to the number of animals censored at each day-of-life. Before day 5306, the distributions of censorship events and death events are depicted as histograms (purple and pink, respectively), in either 100 day or 250 day bins. There were a total of 444 death events and 2803 censorship events after day 183 and prior to day 5306. (**B**) Mortality-hazard estimates (solid green) with 95% confidence intervals (dotted green) across the observed naked mole-rat lifespan. Estimates were calculated across time intervals of increasing size (demarked by grey lines) to compensate for decreasing accuracy-per-unit-time as the population size decreased. The number of observed deaths per bin is indicated in grey.

DOI: https://doi.org/10.7554/eLife.31157.002

For our first repeat analysis, we included 61 animals for whom data was only recorded at month-resolution (blue, *Figure 2A and B*); for our second, we included 139 more animals with lower-resolution data or other modalities of ambiguity (yellow); for our third, we added 69 animals for which birth was recorded as occurring either before or after a date, month, or year (cyan, *Figure 2A and B*; see

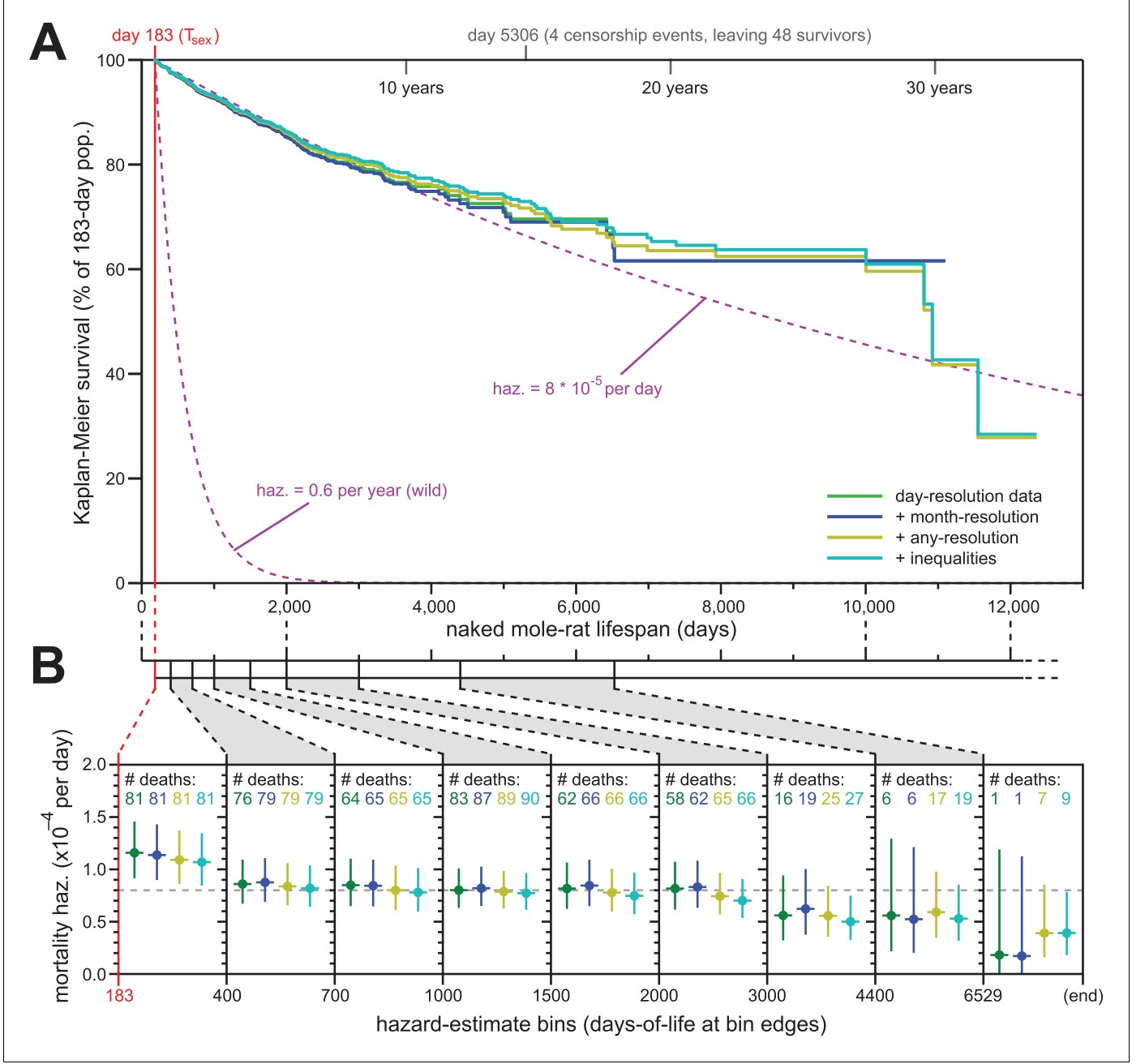

**Figure 2.** Addition of previously excluded, low-resolution lifespan data did not modify the apparent lifespan demographics of *H. glaber*. (**A**) Repeated calculation of Kaplan-Meier survival, using the original data from *Figure 1* (green) and iteratively adding month-resolution data (navy), less-than-month-resolution data (yellow), and inequality-derived data (cyan; see Materials and methods for further descriptions). Expected survivals from $T_{sex}$ (purple) using either our consistent estimate for mortality hazard ($8 * 10^{-5}$ per day) or an estimate from Hochberg (*Hochberg et al., 2016*) for *H. glaber* survival in the wild (0.6 per year). (**B**) Hazard estimates across each of the lifespan bins from *Figure 1B*, calculated for each of the expanded data sets, as described and colored in panel (**A**). Vertical bars indicate 95% confidence intervals. The number of observed deaths per bin is indicated at the top of each panel, colored according to the data set. Dotted grey line indicates $8 * 10^{-5}$ deaths/day.

DOI: https://doi.org/10.7554/eLife.31157.003

Materials and methods for details). The inclusion of less-reliable data did not modify the properties of the overall Kaplan-Meier survival profile (*Figure 2A*). Though inclusion of very-low-resolution data added some deaths to the end of the survival curve (yellow and cyan), those deaths continued to

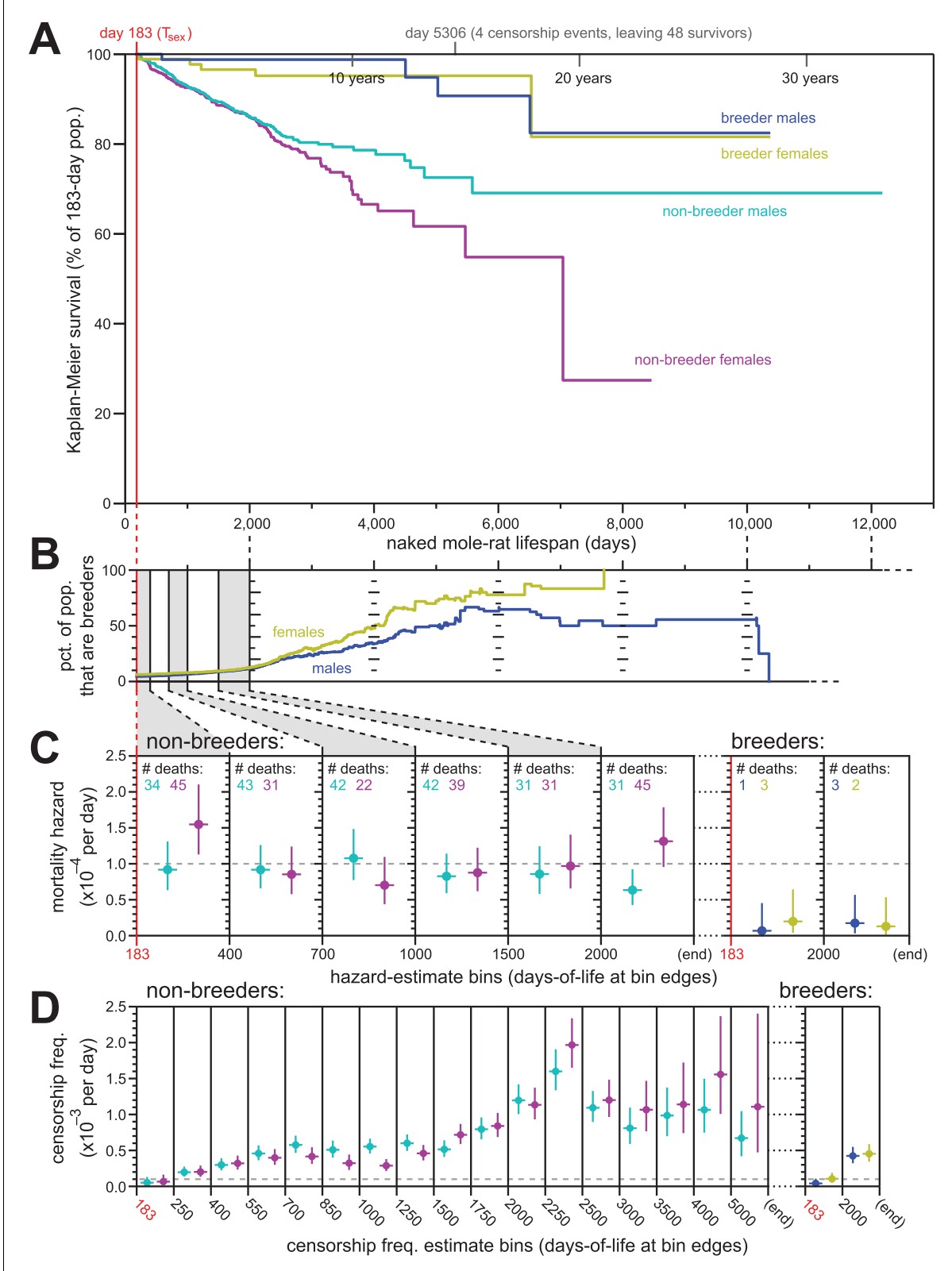

**Figure 3.** Breeding and non-breeding males and females exhibited non-increasing mortal hazard as a function of age. (**A**) Repeated calculation of Kaplan-Meier survival, splitting the data from **Figure 1** into four categories: male breeders (navy), female breeders, (yellow), male non-breeders (cyan), and female non-breeders (purple). (**B**) The percentage of the population recorded as being breeders as a function of age, for females (yellow) and males (navy). (**C**) Hazard estimates for the indicated lifespan bins (x-axis), calculated separately for non-breeders (left) versus breeders (right), and males

*Figure 3 continued on next page*

*Figure 3 continued*

versus females (colored as in panel **A**). Grey dashed line indicates 1/10,000 per day. (**D**) Estimates of the probability of being censored for the indicated lifespan bins (x-axis), displayed separately for non-breeders versus breeders, males versus females, as in panel (**C**). Grey dashed line again indicates 1/10,000 per day.

DOI: https://doi.org/10.7554/eLife.31157.004

closely reflect the expected survival profiles for our estimated constant hazard across the rest of the lifespan (*Figure 2A*, purple, showing our consistent estimate of hazard: $8 * 10^{-5}$ deaths/day).

Likewise, the addition of low-resolution data did little to modify our hazard estimates across longitudinal stretches of lifespan (*Figure 2B*). In fact, the addition of low-resolution data reduced the span of the 95% confidence intervals for the later estimates, adding confidence to our assertion that mortality hazard remains consistent even beyond 6529 days (18 years). While it was tempting to speculate that hazard may even drop at late ages, our confidence intervals only supported such a conclusion when inequality-derived data were included.

## Non-increasing mortality hazard was exhibited by breeders and non-breeders of both sexes

The analyses described above were performed on our full population of animals. Sub-populations defined by sex and breeding-status were analyzed individually for Kaplan-Meier survival (*Figure 3A*). There was a notable survival difference between breeding versus non-breeding status, with breeders enjoying higher survival than non-breeders. However, survival did not observably differ between male versus female breeders. Likewise for non-breeders, survival was similar between the sexes at least until 2000 days of life: after that, the survival curves separated, with males enjoying greater survival than females. While potentially a reflection of biology, we considered an alternative, technical hypothesis that reflected the nature of historical record-keeping for our data: incomplete cataloging of breeder status among males.

Breeder-status was recorded for the founding male and female of each colony. The social hierarchy of naked mole-rat colonies requires that there be only one breeding female per colony (the queen), both in the wild and in captivity (*Jarvis, 1981*). In our records, the founding female of a colony was always demarcated as a breeder; in the event of her death, the subsequent breeder was identified and recorded based on conspicuous morphological changes. It was therefore exceedingly difficult for a female to attain queen status without it being recorded. In the case of males, 1–3 individuals per colony can contribute to breeding. Morphologically, these breeding males are difficult to discern in a non-invasive manner (*Jarvis, 1991*). In our records, breeder-status was defined as a property of founders, not through observations of who was mating with the queen. We therefore must assume that some of the male breeders in our collection – the non-founders – were not labelled as breeding males, and have been analyzed along with the non-breeding males.

The ratio of breeders versus non-breeders was uneven across our lifespan dataset, with the fraction of the population that were breeders increasing as a function of age (*Figure 3B*). Differential survival made only a minor contribution to this difference: the mortal hazards of breeders was ~5–10 fold lower than that of non-breeders (*Figure 3C*). In neither case did the mortal hazard increase with age; analysis was not carried as far into lifespan as in *Figure 1* due to diminished statistical power, but no increase was observable at least until 2000 days of life for both breeding and non-breeding males and females. For female non-breeders, the final hazard estimate (post day-2000) was higher than observed at other earlier time intervals. But as shown by the hazard confidence intervals and reinforced by the simulations for constant-hazard described below, this datum did not discount the relevance of a constant-hazard-with-age demographic model.

In contrast to mortal hazard, censorship probability varied substantially as a function of breeding status and age (*Figure 3D*). This was anticipated from the history for this collection. In particular, the sharp rise of censorship frequency in the 2250-to-2500-day bin (~6.5 years; *Figure 3D*) coincided to the massive expansion of the collection during the years 2007–2011, peaking in 2010. Of the 665 born in 2010 that entered the study population, 52% were still alive and present in the collection when data was compiled six years later, resulting in a large number of censorships at that age (see Supplementary file for a full accounting). Further, this population has been the subject of many

published studies that required the sacrificing of healthy animals in order to collect biological samples for analysis – a censorable endpoint ('killed for research', or 'KFR'; see Materials and methods). The relatively high value of breeders versus non-breeders to the maintenance of the collection has also resulted in a higher frequency of 'KFR' for non-breeders, contributing to the higher overall censorship rates for non-breeders (*Figure 3D*). While censorship rates do not directly influence the results of Kaplan-Meier survival estimation, they do affect the distribution of statistical power for those estimates across lifespan. Below, we explicitly modeled the peculiar censorship history of our collection when using simulations to assess the statistical robustness of our conclusions.

## Simulations supported the relevance of a non-hazard-increasing mortality model to our full historical data set

Based on our knowledge of naked mole-rat social structure, we hypothesized that the deviation in survival between male versus female non-breeders in late life (post-2000-days) was the result of some undocumented breeders being included in the male, but not the female, non-breeder population. To test this hypothesis, we ran simulations for the survival of virtual populations, with censorship structured according to the true history of our collection and under the assumption of mortal hazard being constant with age (*Figure 4*).

The parameters of our survival models for breeders and to non-breeders are outlined in *Figure 4A*. For each model, we assigned age-specific probabilities for censorship and death, starting at day 183 (i.e. $T_{sex}$). The source data for each model parameter is indicated: we defaulted to the values from females given our higher confidence in the comprehensively-accurate labelling of breeders versus non-breeders for that population (purple values). In cases of low statistical power, for which we did not suspect sex to have a true consequence on the estimate, we used a value derived from the combined-sex population (green values). For mortality hazard, we assumed there to be no increase with age, but we included the elevated hazard from the youngest age group that was observed in *Figures 1B*, *2B* and *3B* for females. The exact parameters for our models are also provided in *Supplementary file 2*.

For females, 100 simulations of 1374 non-breeders (the actual number of non-breeders at day 183) yielded results that were statistically consistent with the data from which the parameters were derived (*Figure 4B*). Similarly, for mixed populations of 1374 non-breeders and 91 breeders, 100 simulations yielded results that were statistically consistent with the actual Kaplan-Meier survival for all females (*Figure 4C*).

In contrast to female non-breeders, the simulations for 100 populations of male non-breeders did not match the true observed behavior (*Figure 4D*). However, simulations for the same population but with 175 of the 1747 non-breeders modeled as breeders yielded behavior that was highly consistent with the true observed behavior (*Figure 4E*). There were an additional 83 male breeders labeled as such in the full collection; our estimate of 175 breeders un-labeled in the non-breeder category implied the average true number of male breeders per colony to be approximately three. This was in accord with the estimated number of breeding males in natural naked mole-rat colonies: 1–3 (*Jarvis, 1991*).

## Unlike other mammals, naked mole-rat mortality hazard failed to increase even at ages far beyond that of reproductive maturity

Naked mole-rat mortal hazard failed to increase with age (*Figure 5A*; re-plotted at even time intervals of 200 days). In contrast, other mammalian mortality hazards typically increase exponentially as a function of age after sexual maturity. When we plotted the age-specific hazard of laboratory mice (*Mus musculus*) using the raw data for the control mice cohort from the published paper of Miller (*Miller et al., 2014*), its exponential (i.e., Gompertzian) nature manifested before 800 days-of-life~55% of maximum lifespan (*Figure 5B*). One consequence of Gompertzian hazard is truncation of the latter half of the lifespan distribution. For mice, the time from $T_{sex}$ (42 days post-natal; (*Tacutu et al., 2013*) to median post-$T_{sex}$ lifespan (847 days) was 805 days. The time from median lifespan to maximal lifespan (defined here as the 95th percentile of post-$T_{sex}$ lifespans: 1,141 days for mice) was only 294 days. The latter half of the survival curve was therefore only 37% of the time from median lifespan to maximal lifespan. Gompertzian hazard and this consequence of it were also apparent for humans (*Homo sapiens*; *Figure 5C*) and horses (*Equus ferus caballus*; *Figure 5D*). For

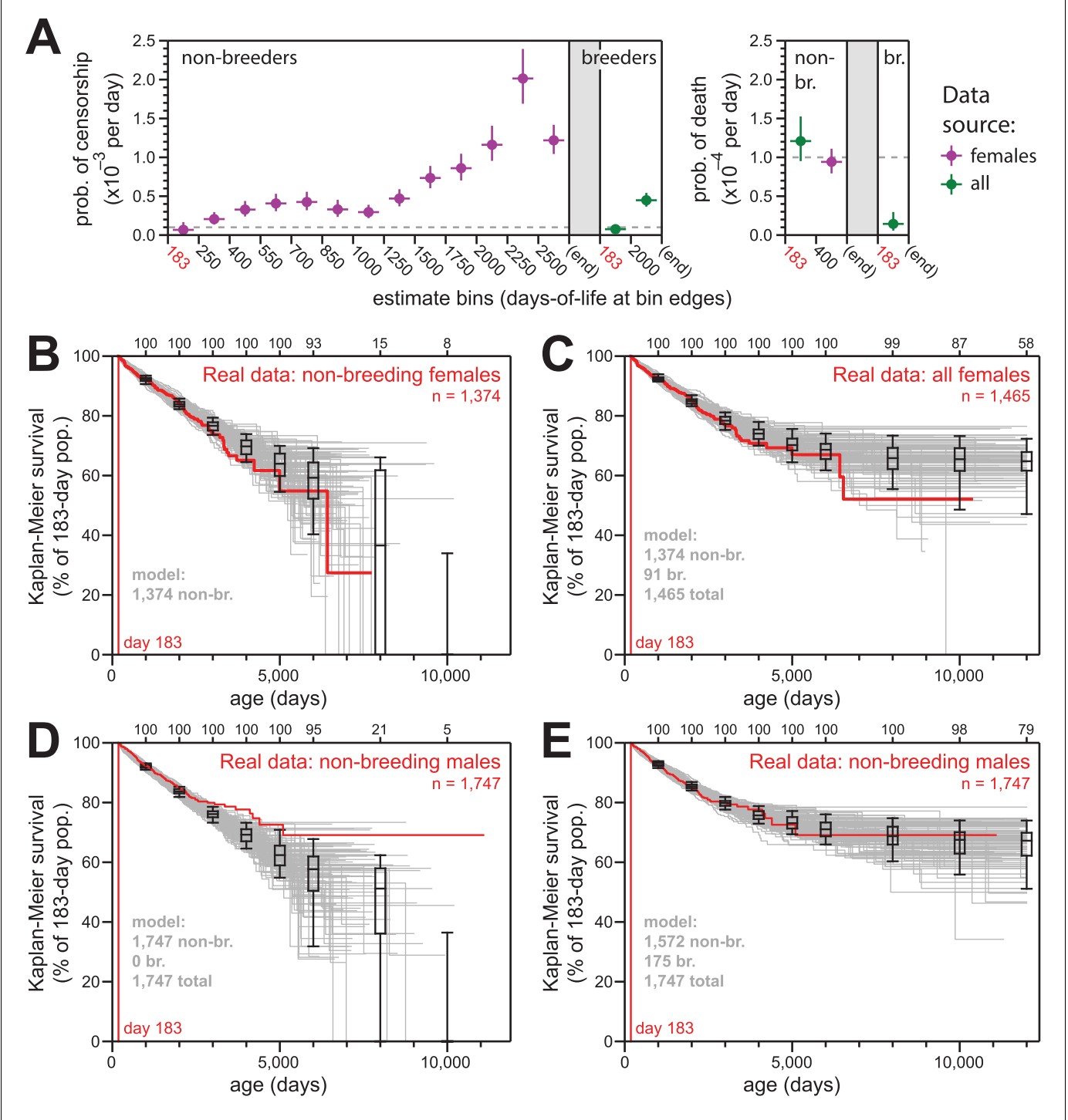

**Figure 4.** Simulations of naked mole-rat survival, run assuming constant mortality hazard, re-capitulated the observed data when breeders and non-breeders were appropriately balanced. (**A**) The model applied here: per-day probabilities of censorship and death for non-breeders and breeders. Separate probabilities were assigned to each age bin indicated on the x-axes. Colors indicate the source data for the estimate used in the model (purple for females; green for both sexes), and error bars indicate 95% confidence intervals for the estimates used to build the model. (**B**) Kaplan-Meier plots for 100 simulations of populations with 1374 non-breeders (grey) versus the true plot for the 1374 non-breeding females (red). Box-and-whiskers indicate the median, quartile, and 5th/95th percentile survival values for simulated populations that had not terminated due to censorship (i.e. 0%-survival populations were included); number of included simulations is indicated along the top axis. (**C**) Kaplan-Meier plots for 100 simulations of populations with 1374 non-breeders and 91 breeders (grey) versus the true plot for the 1374 non-breeding and 91 breeding females (red). Box-and-whiskers as in panel (**B**). (**D**) Kaplan-Meier plots for 100 simulations of populations with 1747 non-breeders (grey) versus the true plot for the 1747

*Figure 4 continued on next page*

*Figure 4 continued*

recorded-as-non-breeding males (red). Box-and-whiskers as in panel (**B**). (**E**) Kaplan-Meier plots for 100 simulations of populations with 1747 individuals, split into 1572 non-breeders and 175 breeders (grey) versus the true plot for the 1747 recorded-as-non-breeding males (red). Box-and-whiskers as in panel (**B**).

DOI: https://doi.org/10.7554/eLife.31157.005

human females, with $T_{sex}$ defined as 16 years (mean age-at-menarche of >12.5 years, plus three years to 50% ovulatory-vs-anovulatory cycle frequency (***Apter, 1980***; ***Anderson et al., 2003***); post-$T_{sex}$ median lifespan was 77.8 years, and post-$T_{sex}$ maximal (95th percentile) lifespan was 96.5 years. The $T_{sex}$-to-median-lifespan time was therefore 61.8 years, and the median-to-maximal-lifespan time was 18.7 years: the latter was 30% of the former. For horses, the 95th percentile of post-$T_{sex}$ survival

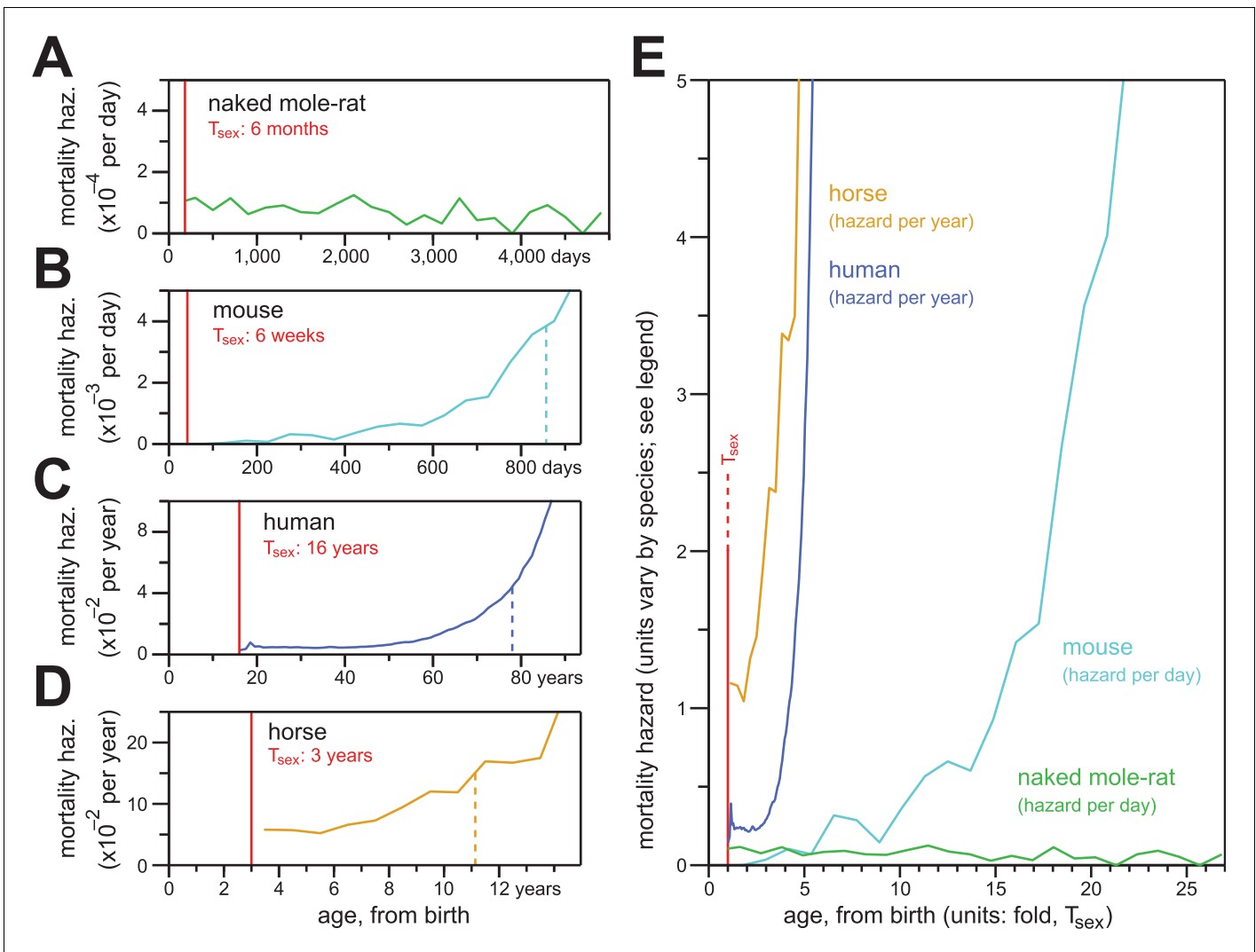

**Figure 5.** In contrast to the mortality hazards of other mammals, which increased with chronological age, the mortality hazard of naked mole-rats remained constant. (**A–D**) Age-specific mortality hazard for naked mole-rats (**A**; green), calculated for 200 day intervals; for mice (**B**; cyan), calculated for 50 day intervals using the control-mouse survival data reported by Miller (***Miller et al., 2014***); for human females (**C**; navy), by year, as reported for the 1900 birth cohort by Bell and Miller (***Bell and Miller, 2005***); and for horses (**D**; orange), by year, calculated from life insurance tables reported by Valgren (***Valgren, 1945***). Calculations begin after $T_{sex}$ for each species (red). (**E**) Hazard plots from panels A-D, re-scaled on the x-axis to the time it takes each organism to reach $T_{sex}$ from birth.

DOI: https://doi.org/10.7554/eLife.31157.006

was not achieved across the insurance tables from Valgren (*Valgren, 1945*). But with $T_{sex}$ defined as three years (*Tacutu et al., 2013*), post-$T_{sex}$ median lifespan was 11.1 years, and only 14.7% of the post-$T_{sex}$ population remained at 16 years. When compared to these other mammals, naked mole-rats distinguished themselves qualitatively and quantitatively by failing to exhibit an increase in mortality hazard even at ages 25-fold greater than the age of sexual maturity.

## Discussion

Aging, defined as the inevitable physiological decline over the lifespan of an organism that weakens homeostasis and increases vulnerability to environmental challenge, is typically observable in mammals in the form of an increase in mortality hazard as organisms get older. A decrease in the rise-rate of hazard has been reported in multiple studies involving experimental interventions that extend lifespan (e.g., caloric restriction), and is consistently interpreted as a slowing of aging itself (*Fok et al., 2014*; *Yen and Mobbs, 2008*; *Simons et al., 2013*). By that metric, 'aging' was not observed at all in our analysis of naked mole-rat mortality, regardless of sex or breeding status. This apparent lack-of-aging agrees thematically with a large body of literature documenting negligible systems-based senescence in naked mole-rats (*Edrey et al., 2011*; *Buffenstein, 2008*). It places this species in a unique demographic position among mammals: as mammals get older, even long-lived species conform to Gompertzian laws of mortality (*Jones et al., 2014*; *Cohen, 2017*).

Delayed (as opposed to non-existent) Gompertzian aging possibly could also explain our observations. The non-increasing hazard could have simply reflected the Makeham component of the Gompertz-Makeham law, a background level of extrinsic hazard that obscures Gompertzian intrinsic hazard early in life. Such a model describes human mortality until age ~40 years (*Bell and Miller, 2005*) (*Figure 5C*) and horse mortality until age ~6 years (*Figure 5D*). We cannot reject the possibility of an as-yet-undetectable Gompertzian component to naked mole-rat mortality. And in fact, this alternative model is not formally falsifiable: any observational extension of constant hazard would simply delay the earliest point at which Gompertzian hazard might take effect. However, given the extremely low baseline hazard rate of 1/10,000 per day for non-breeders and ~1/100,000 for breeders, it seems unlikely that the Gompertzian hazard is being overwhelmed. But even under a delayed-aging model, naked mole-rats profoundly distinguish themselves from other mammals: in this context, through an unprecedentedly-long adulthood prior to the emergence of Gompertzian hazard.

Indeed, naked mole-rat mortality was less consistent with a typical Gompertz model of hazard than it was with an exponential decay model, with a constant probability of decay (death) defining a species' half-life. For naked mole-rats, even the high estimate for per-day hazard of ~1/10,000 that was obtained for non-breeders (*Figure 3A*) would correspond to a half-life (i.e., median lifespan) of ~6900 days (19 years). Under an exponential decay model, the concept of a 'maximal lifespan' loses relevance: at no point does mortality hazard grow into an insurmountable obstacle. One could try and force the concept of 'maximal lifespan' onto an exponentially-decaying species by defining it in terms of a high percentile of lifespan (say, 95%) – and by that definition, the maximal lifespan of the naked mole-rat would be very long indeed! But that definition ignores the implicit meaning of the term: as an age beyond which biological wear and damage has become insurmountable.

Demographic aging is defined by the increase of intrinsic hazard with age (*Gavrilov and Gavrilova, 2015*). Our analyses of captive naked mole-rats confidently revealed a lack of demographic aging up to at least 4400 days of life (~12 years) and strongly suggested no demographic aging far beyond that age (*Figure 1 and 2*). That twelve year stretch of time can be considered from several perspectives. In *Figure 5*, it is considered from the perspective of $T_{sex}$, that is the minimum lifespan allowable for an organism to reproduce. Most mammals live beyond that age, as is required to provide progeny with support and nurturing (*Lee, 2003*), but show signs of demographic aging within a few fold of $T_{sex}$; from this perspective, the naked mole-rat is exceptional, showing no signs of demographic aging many dozens-of-fold beyond $T_{sex}$ (*Figure 5*). Twelve years can also be considered in the context of the expected total lifespan of a naked mole-rat based on its body size (*de Magalhães et al., 2007b*): this 35-gram rodent would be expected to live up to six years, but instead has not shown the first sign of demographic aging at twice that age. Twelve years could finally be considered from the perspective of natural lifespan in the wild, which is estimated to be 2–3 years for the naked mole-rat (*Hochberg et al., 2016*).

Senescence is difficult to observe in wild rodent populations (*David and Jarvis, 1985*; *Krebs et al., 1995*; *Krebs et al., 1973*; *Slade, 1995*) with most rodents living less than a year (*Stueck and Barrett, 1978*) Nonetheless, senescence is easily observed for those rodent species (rats and mice) in captivity, not far beyond their typical natural lifespans (*Figure 5B*; [*Gavrilova and Gavrilov, 2015*]). For the naked mole-rat, demographic aging is absent 4-to-6-fold beyond their natural lifespans.

In summary, naked mole-rats fail to exhibit any evidence for intrinsic hazard contributing to their mortality, even at ages far beyond their expected maximum lifespans based on multiple other aspects of their biology or their full lifespans observed in the wild. In doing so, they establish themselves as a non-aging mammal.

## Materials and methods

The progenitors of the naked mole-rats used in this study were collected by Jennifer Jarvis, Kate Davies, Mike Griffin and Rochelle Buffenstein in Kenya in 1980 and have been housed and cared for at several institutions in South Africa (University of Cape Town and The Medical School of the University of the Witwatersrand) and the USA (City College of New York, University of Texas Health Center at San Antonio and the Buck Institute). Naked mole-rats were housed in multi-chambered plexiglass burrow systems in rooms maintained at 28–30°C and 30–50% relative humidity, in keeping with climatic conditions in their native habitat. The animals were fed *ad-libitum* with fruit and vegetables (bananas, apples, oranges, butternut squash, red bell pepper, romaine lettuce, cucumber, green beans, corn, carrots and red garnet yams) and supplemented with a high protein and vitamin enriched cereal (Pronutro, South Africa).

At 90 days of age, individual mole-rats were sexed, weighed and microchipped, using 12 mm Avid MUSICC microchips, and thereby assigned a unique 9-digit identifier. The microchips were implanted just underneath the dorsal skin in non-anesthetized animals. Thereafter, all animals were weighed regularly at 3 month intervals.

Animal welfare checks were conducted daily. Animals deaths observed during these checks were recorded, the individuals identified from their microchips and necropsies performed. Records were aggregated into three spreadsheets. For each spreadsheet, the dates-of-censorship for naked mole-rats currently alive and in the collection were set to the date of completion for the spreadsheet in which they were recorded (April 14, 2016; April 20, 2016; and May 17, 2016). Those three spreadsheets are provided as a single file (*Supplementary file 1*), with the date of compilation indicated. Censorship status was additionally applied to 'dead' animals labeled as 'SOLD/GIVEN AWAY' (409 animals) or 'KFR' (killed for research; 867 animals). Lifespan was calculated in days as the difference between the two specified dates, using the python 'datetime' module. Animals with dates of birth or death/censorship with less-than-single-day-resolution (e.g., month- or year-of-birth available) were excluded from the initial analysis and from analyses based on sex and breeding status, but were included in the 'expanded' analyses from *Figure 2*. Animal-by-animal data is provided in *Supplementary file 1* (see Legend for details).

For our first expanded analysis, we included data that was only recorded at month-resolution (blue, *Figure 2A and B*). We used the 15th day of each month as our midpoint 'guess'. Month-of-birth was reported for 38 individuals. For an additional 22, approximate month-of-birth was recorded: we treated these the same as unambiguously recorded month-of-birth. For one individual, month-of-death was reported. Of those 61 animals added to our dataset, 19 died and 42 were right-censored.

For our second expanded analysis, in addition to the 61 animals from above, we included 139 more animals with lower-resolution data or other modalities of ambiguity (yellow, *Figure 2A and B*). For two animals, year and season of birth was recorded, in both cases as 'Fall'. We used November 7 as the seasonal mid-point of Fall. For 36 individuals, ranges of possible dates were given, with the endpoints given at the resolution of years for 24 of those. For each range, we took the middle date of the full timespan described by the range. The year-of-birth alone was recorded for 40 individuals: we used July 2 as the year mid-point. For an additional 57 animals, an approximate year was recorded: we treated those in the same manner as specific years. Two optional dates were provided for four individuals: we treated each of these as a range, using the mid-point between the two options. Of those 139 animals added to our dataset, 28 died and 111 were right-censored. Including

the 61 added above (200 animals added in total versus the original analysis), 47 died and 153 were right-censored.

For our third expanded analysis, in addition to the 200 animals from above, we added 69 animals for which birth was recorded as occurring either before or after a date, month, or year (cyan, *Figure 2A and B*). This is the worst-case scenario in terms of data reliability: the indicated time-of-birth was un-bounded in one direction (or bounded by the present), making it impossible to assign an average of the possibilities as was the case for ranges. Nonetheless, to include these animals in the analysis, we used the closest date satisfying each inequality (e.g. treating '>1997' as January 1, 1998). Of those 69 animals added to our dataset, 8 died and 61 were right-censored. Including the 200 added above (269 animals added total versus the original analysis), 55 died and 214 were right-censored.

For *Figures 1–4*, survival was calculated using the Kaplan-Meier method (*Kaplan and Meier, 1958*). For *Figures 1–3*, mortality hazard for each block of days/group of animals was calculated as the number of observed death events (i.e., number of days on which a naked mole-rat was observed to have died) divided by the total number of days that naked mole-rats were under observation (for individuals who were alive across an entire interval: the number of days in the block; for individuals who died or were censored during a block: the number of days in the block prior to and including the death/censorship event). Confidence intervals were calculated using the Wilson score interval with continuity correction (*Newcombe, 1998*).

For naked mole-rats in *Figure 5A*, hazards were calculated as described above across non-over-lapping 200 day windows, starting at day 200. Hazard was also calculated across the shorter interval between the day of reproductive maturity and the beginning of the first full window (days 183–199). For mice in *Figure 5B*, the same method was applied to non-overlapping 50 day windows starting at day 50, with hazard additionally calculated for the partial window starting at reproductive maturity (days 42–49). Mouse data was the combined male-and-female control-mouse survival data reported by Miller (*Miller et al., 2014*), collected for the NIA Intervention Testing Program (*Miller et al., 2007*; *Nadon et al., 2008*). Raw data for control mice used in *Figure 5* (*Miller et al., 2014*) was provided by the authors.

A model for survival for breeders versus non-breeders was constructed by calculating the hazards for death or censorship using data from either females-only or males-and-females, and for windows of age, as-specified in *Figure 4A*. Mortality hazard was calculated as described above; censorship hazard was calculated as described for mortality hazard but reversing the treatment of censorship versus death events.

Lifespan simulations were performed using custom python scripts. For each simulation, an initial population was created with the indicated-in-the-figure number of breeders and non-breeders. For each day starting at day 183 (i.e. $T_{sex}$), for each surviving and uncensored individual, a pseudo-random float in the half-open interval [0.0, 1.0) was generated using the python 'random' module. If that number was less than the probability of censorship given breeding-status and day-of-life, the individual was marked as 'censored' on that day and removed from the population. Otherwise, a second pseudo-random float was generated: if that number was less than the probability of death given breeding-status and day-of-life, the individual was marked as 'died' on that day and removed from the population. The simulation continued until the population had been exhausted or day 12,000 was reached, in which case all surviving individuals were marked as 'censored' on that day.

Parameters of our model are provided in *Supplementary file 2*. The exact demographic data for panels B-E of *Figure 4* (100 simulations per panel) are provided as *Supplementary file 3–6*, respectively. A python script for running additional simulations as described above is provided as Supplemental Python Script.

For human in *Figure 5C*, hazards were plotted as calculated and reported for females of the 1900 birth cohort by Bell and Miller (*Bell and Miller, 2005*). For horse in *Figure 5D*, hazards were calculated using two insurance life-tables from Valgren (*Valgren, 1945*): data from Table 1 and 2 were combined, and hazard estimated as the total dead divided by the total insured for each year-of-life. Median and percentile survivals were calculated through iterative multiplication of age-specific survival (one minus age-specific hazard), starting at $T_{sex}$.

## Acknowledgements

We sincerely thank Hal Barron, David Botstein, Cynthia Kenyon and Patrick Gibney at Calico Life Sciences for critique of this manuscript. We thank the numerous animal attendants who have diligently cared for the naked mole-rats over the past three decades. Richard Miller (University of Michigan) kindly provided access to the raw data published in (*Miller et al., 2014*), used here in *Figure 5B*. Calico Life Sciences kindly funded this study.

## Additional information

### Competing interests

J Graham Ruby, Megan Smith, Rochelle Buffenstein: The research was funded by Calico Life Sciences LLC, where all authors were employees at the time the study was conducted. The authors declare no other competing financial interests.

### Funding

This study was funded by Calico Life Sciences LLC.

### Author contributions

J Graham Ruby, Conceptualization, Data curation, Formal analysis, Visualization, Writing—original draft, Writing—review and editing; Megan Smith, Data curation, Investigation; Rochelle Buffenstein, Conceptualization, Resources, Data curation, Supervision, Investigation, Methodology, Writing—original draft, Project administration, Writing—review and editing

### Author ORCIDs

Rochelle Buffenstein (D) https://orcid.org/0000-0003-3285-8311

### Ethics

Animal experimentation: All of the animals were handled according to approved institutional animal care and use committee (IACUC) protocols, most recently of the Buck Institute (A10138). IACUC approval for housing this colony of naked mole-rats was also obtained from all institutions at which this collection of naked mole-rats were housed over this three decade long study.

### Decision letter and Author response

Decision letter https://doi.org/10.7554/eLife.31157.016
Author response https://doi.org/10.7554/eLife.31157.017

## Additional files

### Supplementary files

• Source code 1. Supplemental Python Script. A simulator for breeder/non-breeder population survival simulation, implementing the method described for *Figure 4* and with output formatted as in *Supplementary file 3–6*. Run the script using python (recommended: Python 2.7.13) with no additional arguments for user-level documentation. Read the script (provided as a text document) for developer-level documentation.
DOI: https://doi.org/10.7554/eLife.31157.007

• Supplementary file 1. Animal-by-animal lifespan and annotation data. This is a comma-separated text file with 11 columns. Each row in the file contains data for one animal. Column 1 ('IsDead') indicates whether the animal was dead or alive at the time these data were compiled. Column 2 ('AnimalID') is a unique identifier for each animal (to facilitate reference/discussion). Column 3 ('BirthDate') is the date-of-birth according to our records (varying degrees of resolution: month/day/year in the ideal case). Column 4 ('Sex') indicates both the sex of the animal ('M'/'F' for male/female) and breeding status (addition of a 'B' when an individual was known to be a breeder), as documented in our historical records. Column 5 ('Notes') include information on the death or censorship

event (classified as described in the previous paragraph). Column 6 ('DeathDate') indicates, for dead/censored animals, the date of that event. It is empty if the animal were still alive at the time of data compilation. Column 7 ('DataDate') indicates the date on which data for that animal was compiled. If the animal were still alive, that date is used as the date-of-censorship. Column 8 ('ValidBirth') indicates whether the birth was single-day-resolution, that is sufficient for inclusion in the analysis set for *Figure 1* ('Y'/'N' for yes/no). Column 9 ('ValidDeathCen') gives the same indication for the date of death or censorship. Column 10 ('Censored') indicates the classification of death-versus-censorship ('Y' indicates censorship). Column 11 ('Lifespan(days)") gives the number of days of life for all animals with 'Y' values (indicating single-day-resolution) in both columns 8 and 9.

DOI: https://doi.org/10.7554/eLife.31157.008

• Supplementary file 2. Parameters for demographic simulations. This is a comma-separated text file with 5 columns. Column 1 ('DeathOrCen') indicates if the given row provides hazard for either death ('D') or censorship ('C') events. Column 2 ('Breeder') indicates whether the row provides hazard for breeders (yes; 'Y') or non-breeders (no; 'N'). Column 3 ('Start') indicates the first day to which the hazard from that row should be applied. Column 4 ('Finish') indicates the day on which the hazard from that row should stop being applied (or '(end)' if the value should be applied indefinitely). Column 5 ('Hazard/day') gives the per-day probability (hazard) of the event described in column 1 occurring for the animal described in column 2.

DOI: https://doi.org/10.7554/eLife.31157.009

• Supplementary file 3. Simulations from *Figure 4B*. These simulations were run as described in Methods, with a starting population of 1374 non-breeders. One hundred simulations were run. This is a comma-separated text file with 4 columns. Each row in the file contains data for one simulated animal. Column 1 ('Sim') indicates which of the 100 simulations the animal was a part. Column 2 ('Br') indicates breeding status ('Y' for breeders, 'N' for non-breeders). Column 3 ('Ls') indicates the lifespan of the animal, in days. Column 4 ('Cen') indicates whether the animal's lifespan ended with censorship or death ('Y' for censorship, 'N' for death).

DOI: https://doi.org/10.7554/eLife.31157.010

• Supplementary file 4. Simulations from *Figure 4C*. These simulations were run as described in Methods, with a starting population of 91 breeders and 1374 non-breeders. One hundred simulations were run. This is a comma-separated text file, formatted as described for *Supplementary file 3*.

DOI: https://doi.org/10.7554/eLife.31157.011

• Supplementary file 5. Simulations from *Figure 4D*. These simulations were run as described in Methods, with a starting population of 1747 non-breeders. One hundred simulations were run. This is a comma-separated text file, formatted as described for *Supplementary file 3*.

DOI: https://doi.org/10.7554/eLife.31157.012

• Supplementary file 6. Simulations from *Figure 4E*. These simulations were run as described in Methods, with a starting population of 175 breeders and 1572 non-breeders. One hundred simulations were run. This is a comma-separated text file, formatted as described for *Supplementary file 3*.

DOI: https://doi.org/10.7554/eLife.31157.013

• Transparent reporting form

DOI: https://doi.org/10.7554/eLife.31157.014

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
