## [Decision Letter]

Thank you for submitting your article "Naked Mole-Rat Mortality Rates Defy Gompertzian Laws By Not Increasing With Age" for consideration by *eLife*. Your article has been reviewed by 3 peer reviewers, and the evaluation has been overseen by Michael Rose as the Reviewing Editor, and Patricia Wittkopp as the Senior Editor. The reviewers have opted to remain anonymous.

The reviewers have discussed the reviews with one another and the Reviewing Editor has drafted this decision to help you prepare a revised submission.

Summary:

The naked mole rat stands out as an example of a small non-flying mammal that is exceptionally long-lived. This article gathers together extensive records of the life histories of thousands of naked mole rats for the first time. Understanding the patterns revealed by these data will be fundamental to the development of a deeper understanding of mammalian aging.

Essential revisions:

After discussion among the reviewers, we agree that the manuscript presents important data that should be published if (i) an appropriate demographic analysis is done, and (ii) the scientific inferences drawn from any such analysis are appropriately couched. Our suggestions for the major revision of the manuscript thus fall under two headings. The following attempts to highlight our main recommendations. However, careful attention to the specific comments of the individual reviews is also required.

1) Extensive Disaggregated Demographic Re-Analysis

A re-analysis based on sex is a minimum requirement. The analysis of these data sorting out the reproductive and non-reproductive individuals will be complicated because there is no fixed age at which they become reproductive, but should also be done. While the low and relatively constant mortality rate early in life is interesting, it is doubtful that there are enough data to support the headline that these animals don't age. While it is reasonable to conclude that up to day 4400 the hazard rates seem roughly constant, we can't say much after this age. However, if these animals indeed live up to 103 years, then 4400 days is only about 12% of the lifespan. Thus it is unreasonable to conclude that this animal does not age when such a small portion of the lifespan has been studied. Nonetheless, the distinctively long lifespans of these small mammals are of great scientific interest.

2) Appropriately Focused Scientific Inference

As the work of R. Lee makes clear, Hamilton's demographic theory for the evolution of aging is actually a special case of a more general theory of life-history evolution that incorporates both demography and kin relatedness. Hamilton's 1966 theory is patently inadequate, from first principles, to explain the evolution of mortality rates in eusocial species like honeybees, termites, or indeed the naked mole rat. Therefore, using the naked mole rat to test that 1966 theory is highly inappropriate.

The Corresponding Editor looks forward to receiving a greatly revised manuscript.

*Reviewer #1:*

These are very important data. But there are substantial problems with this study that need to be addressed in a major revision.

1) While I have attempted to read the raw data file, I have only had limited success with understanding what is in it. This, together with a lack of explanation of the reproductive status in data from the only "eusocial" vertebrate, create a major problem of interpretation of the results. Specifically, while the discussion of the data mentions both reproductive and non-reproductive individuals anecdotally, there is a major difficulty with analysis of these demographic data: which of these individuals is a "pre-reproductive juvenile" and which individuals are fully reproductive adults? If many of these individuals are such "juveniles," then they are not expected to exhibit Hamiltonian aging, in terms of Hamiltonian force-of-natural-selection aging. On the other hand, the "kings" and "queens" of the analogous eusocial termite species can live far longer than their "juvenile" non-reproductive workers. Perhaps the same thing is true of dominant reproductives in this mammalian species?

Thus there is a major problem of protracted heterogeneity in the cohort, with some individuals physiologically "juvenile" and other individuals physiologically adult in every sense. Without that heterogeneity teased apart, it is a long-standing finding of demographic analysis that protracted heterogeneity is expected to obscure underlying within-group demographic aging, so long as the sub-cohort heterogeneity is not unmasked.

2) There is also a substantive problem of scientific inference with the manuscript. As the work of demographers like R. Lee (UC Berkeley) makes clear, Hamilton's demographic theory for the evolution of aging is actually a special case of a more general theory of life-history evolution that incorporates both demography and kin relatedness. [And that theory in turn is a special case of the general Price equation.] For a species like the naked mole rat, whose evolution has evidently been dominated by the effects of kin selection, the relevant theory is the broader one that Lee has developed. Hamilton's 1966 theory is patently inadequate, from first principles, to explain the evolution of mortality rates in eusocial species like honeybees, termites, or indeed the naked mole rat. Therefore, using the naked mole rat to test that 1966 theory is highly inappropriate.

3) Properly disaggregated and parsed analysis of the different kinds of naked mole rat in the cohort data would lead to an analysis which could be usefully applied to the testing of the Lee model for life-history evolution, as well as other models of that general kind. As such, this research is potentially of greater importance than the authors of the manuscript may realize.

*Reviewer #2:*

These findings have high impact. The absence of mortality rate increases at later ages is convincingly shown.

1) Introduction: Cancer cannot be "exceedingly rare" in view of the 5 cases recently reported in 37 necropsies of zoo animals (Taylor et al., J Gerontol 2017).

2) Results: If both sexes are included, show both and examine possible sex differences in mortality.

3) Discussion: Expand comparisons of NMR with other rodent "background mortality" or Gompertz scalar intercept: Fok et al., PLoS One 2014, Simons et al., Aging Cell 2013, Yen et al., Exp Gero 2008.

*Reviewer #3:*

The authors analyze survival data from a large captive population of mole rates, a eusocial mammal in which most females forgo reproduction for at least part of their life. The main thrust of the paper is that estimated hazard rates appear to constant of the early portion of the lifespan for which the authors can obtain estimates and this suggests an organism that does not age. I don't believe the data and analysis support this conclusion. At best these data may support the idea that mortality is roughly constant for the early, say first 12% of the lifespan of these organisms.

In general the authors do a good job describing the population of mole rats they studied however they give a rather sparse treatment of important demographic details. For instance, presumably there are males in the population but there is no indication of how many males and female were in the initial cohort 3,299 used for Figure 1. Presumably the two sexes were pooled but it is not unreasonable to expect different rates of mortality in different sexes and thus to separate out the sexes.

More importantly females themselves are heterogeneous belonging to either a reproducing or non-reproducing class. In this study the authors point out there are at least multiple females that started reproduction before age 3 years and at least one female that did not start reproducing until age 22. The literature is full of examples of the effects of reproduction on survival which in general suggests a decrease in longevity due to reproduction. Thus, the analysis needed to separately compare the mortality of breeding females vs. non-breeding females. Mixing these results may obscure patterns of aging.

It should not be surprising that almost any animal brought into captivity and provided with ample nutrition, freedom from predators and most diseases will live longer than in the wild. Indeed it is under these conditions that intrinsic sources or mortality that make up the aging process can best be measured. While there are apparently rough estimates of these animals living beyond 30 years in nature the authors discover that in captivity their population easily results in individual surviving to this age. Indeed, they in fact estimate that 5% of a cohort may survive to 103 years or longer.

If this is in fact the case we can now examine what portion of the lifespan of this organism has been examined in the present study. Of course the practical problems faced by this study is that (i) these organisms live too long to wait for all of them to die, and (ii) so many individuals are censored that very quickly there are too few survivors to provide accurate estimates of mortality rates. Let's examine Figure 1 in some detail to highlight this problem.

Figure 1 provides the crucial analysis for the authors conclusions. Here the authors have chosen intervals to estimate hazard rates but allowed the intervals to vary in length so as to provide about the same statistical accuracy. We see that basically after 4400 days the confidence intervals on the hazard rates have become so great that only exceptionally large changes in hazard rates would be detectable. Again this is due to the fact that despite a large initial cohort there are very few survivors and thus deaths at this point due to censoring. While it is reasonable to conclude that up to day 4400 the hazard rates seem roughly constant (ignoring the complicating issues I raised above) we can't say much after this age. However, if these animals might indeed live up to 103 years then 4400 days is only about 12% of the lifespan. I think it is unreasonable to conclude that this animal does not age when such a small portion of the lifespan has been studied.

---

## [Author Response]

Essential revisions:After discussion among the reviewers, we agree that the manuscript presents important data that should be published if (i) an appropriate demographic analysis is done, and (ii) the scientific inferences drawn from any such analysis are appropriately couched. Our suggestions for the major revision of the manuscript thus fall under two headings. The following attempts to highlight our main recommendations. However, careful attention to the specific comments of the individual reviews is also required.1) Extensive Disaggregated Demographic Re-AnalysisA re-analysis based on sex is a minimum requirement. The analysis of these data sorting out the reproductive and non-reproductive individuals will be complicated because there is no fixed age at which they become reproductive, but should also be done.

In response to the reviewers’ constructive critique, we have reanalyzed our data based upon sex and breeding status and created a new figure (Figure 3) stratifying the analysis by both sex and breeding status. We also have added a large block of text to the manuscript describing these new results. The reviewers were correct that interpretation of breeders versus non-breeders is complicated. In order to address the caveats of that analysis, we performed a new set of analyses involving computer simulations of population data given mixtures of breeders and non-breeders, which we present in another new figure (Figure 4) and included a corresponding section to the Results. We also provide the actual simulation results used for Figure 4 as Supplementary file 3–Supplementary file 6, and we provide a python script in the supplements that will allow readers to run additional simulations according to our described model. We believe these additional data considerably strengthen this manuscript, further highlighting our conclusion of no evidence of Gompertzian aging.

While the low and relatively constant mortality rate early in life is interesting, it is doubtful that there are enough data to support the headline that these animals don't age. While it is reasonable to conclude that up to day 4400 the hazard rates seem roughly constant, we can't say much after this age. However, if these animals indeed live up to 103 years, then 4400 days is only about 12% of the lifespan. Thus it is unreasonable to conclude that this animal does not age when such a small portion of the lifespan has been studied. Nonetheless, the distinctively long lifespans of these small mammals are of great scientific interest.

We thank all the reviewers for appreciating the distinctively long lifespans of these mouse-sized rodents and for recognizing the novelty and importance of the finding of a constant, stochastic, intrinsic mortality for the extensive period of at least 4400 days, ~ approximately 3000 days longer than the maximum lifespan for mice. However, we feel that we’ve been unsuccessful at communicating the significance of this important finding to you, and we respectfully disagree with your conclusion that 4400 days of non-increasing hazard is insufficient to come to any conclusions about the aging properties of this species. We explain why below:

Here we show for the first time that a small mammal expected to live 6 years (on the basis of allometric assessments involving more than 1000 species: de Maghalaes et al., 2007), has yet to show the *first* signs of demographic aging at (4400 days), i.e., a time period *double* its predicted *maximum* lifespan. Moreover, although we appreciate that all animals live longer in captive environments than they would in the wild, our observation of consistent hazard (i.e., unchanged intrinsic mortality with age or demographic non-aging) to 4400 days extends over *four-fold* longer than the observed lifespan*s* of non-breeders in the wild. Further still, we observed non-demographic-aging to extend many-fold beyond the observed *maximum-lifespan*-to-T_sex_ ratio observed in other mammals. Reemphasizing our most important point: we are highlighting that naked mole-rats, in our care, fail to exhibit any evidence for intrinsic hazard contributing to their mortality, even at ages far beyond their expected maximum lifespans based on other aspects of their biology or their full lifespans observed in the wild.

The concept of “maximum lifespan” logically implies that, at some age, the damage that has accumulated over life has become insurmountable to normal biological function. This results in a breakdown in homeostatic maintenance, a decline in function and an increased susceptibility to poor health and dying. In other words, the concept of “maximum lifespan” implies and *requires* an increase of hazard with age. In our manuscript, we tried to make the point that the concept of “maximum lifespan” is irrelevant to a species with constant hazard, stating that “Under an exponential decay model, this concept loses relevance: at no point does mortality hazard grow into an insurmountable obstacle”. We have tried to enhance the emphasis on this point, and used more text to spell out its implication: that naked mole-rat lifespan demographics are defined by extrinsic rather than intrinsic hazard, far beyond their survival in the wild or their expected survival when compared to mammals that do ultimately succumb to extrinsic hazard.

Demographic aging is defined by the increase of intrinsic hazard with age. Again, our point is that we do not observe any such aging far beyond what should be their full lifespan based on body size, what they need their lifespan to be given their time to reproductive maturity, or what their lifespan actually is in the wild. For those reasons, we feel justified referring to the naked mole-rat as a non-aging species of mammal. As we described above, we feel that “maximum lifespan” is an absurd concept for a non-aging species. Unfortunately, we originally highlighted that absurdity by calculating and presenting the insanely-high value that is obtained if the concept is forced onto the naked mole-rat: over a century. Since reviewer #3 took that as a realistic estimate of typical lifespan, we have removed that estimate from the manuscript so as to keep focus on the defiance of age-associated Gompertizan hazard. We’ve replaced it with a more-lengthy explanation of the irrelevance of the maximal lifespan concept to a non-aging species, and simply pointed out that any attempt to force this irrelevant concept onto a non-aging organism will result in an estimate that is “very long indeed!”.

2) Appropriately Focused Scientific InferenceAs the work of R. Lee makes clear, Hamilton's demographic theory for the evolution of aging is actually a special case of a more general theory of life-history evolution that incorporates both demography and kin relatedness. Hamilton's 1966 theory is patently inadequate, from first principles, to explain the evolution of mortality rates in eusocial species like honeybees, termites, or indeed the naked mole rat. Therefore, using the naked mole rat to test that 1966 theory is highly inappropriate.

We have been convinced by the reviewers and Corresponding Editor that the Hamilton theory is irrelevant and have removed any reference to it from the paper. This does not detract from the key findings of this study.

The Corresponding Editor looks forward to receiving a greatly revised manuscript.Reviewer #1:These are very important data. But there are substantial problems with this study that need to be addressed in a major revision.1) While I have attempted to read the raw data file, I have only had limited success with understanding what is in it.

We have added a paragraph to the Materials and methods that describes the format of Supplementary file 1 and spelling out the meaning of each column of data. It contains all of the raw census data used in the analyses.

This, together with a lack of explanation of the reproductive status in data from the only "eusocial" vertebrate, create a major problem of interpretation of the results. Specifically, while the discussion of the data mentions both reproductive and non-reproductive individuals anecdotally, there is a major difficulty with analysis of these demographic data: which of these individuals is a "pre-reproductive juvenile" and which individuals are fully reproductive adults? If many of these individuals are such "juveniles," then they are not expected to exhibit Hamiltonian aging, in terms of Hamiltonian force-of-natural-selection aging. On the other hand, the "kings" and "queens" of the analogous eusocial termite species can live far longer than their "juvenile" non-reproductive workers. Perhaps the same thing is true of dominant reproductives in this mammalian species?Thus there is a major problem of protracted heterogeneity in the cohort, with some individuals physiologically "juvenile" and other individuals physiologically adult in every sense. Without that heterogeneity teased apart, it is a long-standing finding of demographic analysis that protracted heterogeneity is expected to obscure underlying within-group demographic aging, so long as the sub-cohort heterogeneity is not unmasked.

We agree that including data on breeding status and sex adds another dimension to these findings. As described above, we have added additional analysis, two figures, and extensive text discussing the influence of breeding status and sex on survival and hazard. Moreover, we have added computer simulations of our mortality models. We have also removed all references to Hamilton’s force-of-natural-selection aging.

2) There is also a substantive problem of scientific inference with the manuscript. As the work of demographers like R. Lee (UC Berkeley) makes clear, Hamilton's demographic theory for the evolution of aging is actually a special case of a more general theory of life-history evolution that incorporates both demography and kin relatedness. [And that theory in turn is a special case of the general Price equation.] For a species like the naked mole rat, whose evolution has evidently been dominated by the effects of kin selection, the relevant theory is the broader one that Lee has developed. Hamilton's 1966 theory is patently inadequate, from first principles, to explain the evolution of mortality rates in eusocial species like honeybees, termites, or indeed the naked mole rat. Therefore, using the naked mole rat to test that 1966 theory is highly inappropriate.

We have been convinced by the reviewer that Hamiltonian theory is irrelevant and feel addressing in depth the various life-history evolutionary theories is beyond the scope of this manuscript. We therefore have removed all references to it and other aging theories.

3) Properly disaggregated and parsed analysis of the different kinds of naked mole rat in the cohort data would lead to an analysis which could be usefully applied to the testing of the Lee model for life-history evolution, as well as other models of that general kind. As such, this research is potentially of greater importance than the authors of the manuscript may realize.

We agree that the data likely have relevance to the theory of R. Lee. However, as was the case with Hamilton, the goal of this paper was not to test any evolutionary theory of aging, it was to demonstrate the deviation of *H. glaber* lifespan demographics from those described by Gompertz. We have therefore removed the references to Hamilton and not attempted to replace them with explicit discussions of other evolutionary theories of aging. All of the primary data for our analyses is available to the larger community in our Supplementary file if others wish to test specific evolutionary theories more explicitly.

Reviewer #2:These findings have high impact. The absence of mortality rate increases at later ages is convincingly shown.1) Introduction: Cancer cannot be "exceedingly rare" in view of the 5 cases recently reported in 37 necropsies of zoo animals (Taylor et al., J Gerontol 2017).

We removed the word “exceedingly”, although based upon mouse pathology data it appears that the relative incidence of cancer in naked mole-rats can be described in such terms. We have had only 4 incidences of cancer out of >2500 necropsies. So although we cannot state categorically that naked mole-rats are impervious to cancer, the incidence of cancer relative to that observed in other laboratory based rodent studies remains “rare”.

2) Results: If both sexes are included, show both and examine possible sex differences in mortality.

As discussed above, we’ve created new figures and a new section of text to address this point (along with breeding status).

3) Discussion: Expand comparisons of NMR with other rodent "background mortality" or Gompertz scalar intercept: Fok et al., PLoS One 2014, Simons et al., Aging Cell 2013, Yen et al., Exp Gero 2008.

We have added to both the Introduction and Discussion your suggested with the references from the above (Fok, Simons and Yen) as well as additional papers on rodent lifespan demographic (Davies, Slade, Steuck, Krebs, Gavrilova).

Reviewer #3:The authors analyze survival data from a large captive population of mole rates, a eusocial mammal in which most females forgo reproduction for at least part of their life. The main thrust of the paper is that estimated hazard rates appear to constant of the early portion of the lifespan for which the authors can obtain estimates and this suggests an organism that does not age. I don't believe the data and analysis support this conclusion. At best these data may support the idea that mortality is roughly constant for the early, say first 12% of the lifespan of these organisms.

We respectfully disagree with this comment and realize that we did not clearly explain the point we were trying to get across. By all metrics that one can use for comparison, the mouse-sized naked mole-rat has an extraordinarily long lifespan in captivity and most importantly even at ages far greater than that of any other rodent species still shows no age-associated increase in intrinsic mortality. (Please see comments to the editors).

In general the authors do a good job describing the population of mole rats they studied however they give a rather sparse treatment of important demographic details. For instance, presumably there are males in the population but there is no indication of how many males and female were in the initial cohort 3,299 used for Figure 1. Presumably the two sexes were pooled but it is not unreasonable to expect different rates of mortality in different sexes and thus to separate out the sexes.More importantly females themselves are heterogeneous belonging to either a reproducing or non-reproducing class. In this study the authors point out there are at least multiple females that started reproduction before age 3 years and at least one female that did not start reproducing until age 22. The literature is full of examples of the effects of reproduction on survival which in general suggests a decrease in longevity due to reproduction. Thus, the analysis needed to separately compare the mortality of breeding females vs. non-breeding females. Mixing these results may obscure patterns of aging.

We thank the reviewer for these suggestions; initially we chose not to do this as we felt we could easily distinguish breeding females from subordinate females but not that of breeding and non-breeding males. The breeding female will mate with the male we pair her with but as the colony gets larger, she will also begin to mate with 1-3 of her offspring. As discussed above, we have performed separate analyses on breeding-versus-non-breeding males-versus-females, adding extensive text on these analyses and two additional figures. Sex did not appear to affect demography, though breeding-status did. Breeding females, in defiance of the disposable soma theory of aging, show the most robust constant mortality rates. However, no sub-population presented credible evidence of hazard-increase. As such, our overall conclusions were unaffected.

It should not be surprising that almost any animal brought into captivity and provided with ample nutrition, freedom from predators and most diseases will live longer than in the wild. Indeed it is under these conditions that intrinsic sources or mortality that make up the aging process can best be measured. While there are apparently rough estimates of these animals living beyond 30 years in nature the authors discover that in captivity their population easily results in individual surviving to this age. Indeed, they in fact estimate that 5% of a cohort may survive to 103 years or longer.If this is in fact the case we can now examine what portion of the lifespan of this organism has been examined in the present study. Of course the practical problems faced by this study is that (i) these organisms live too long to wait for all of them to die, and (ii) so many individuals are censored that very quickly there are too few survivors to provide accurate estimates of mortality rates. Let's examine Figure 1 in some detail to highlight this problem.Figure 1 provides the crucial analysis for the authors conclusions. Here the authors have chosen intervals to estimate hazard rates but allowed the intervals to vary in length so as to provide about the same statistical accuracy. We see that basically after 4400 days the confidence intervals on the hazard rates have become so great that only exceptionally large changes in hazard rates would be detectable. Again this is due to the fact that despite a large initial cohort there are very few survivors and thus deaths at this point due to censoring. While it is reasonable to conclude that up to day 4400 the hazard rates seem roughly constant (ignoring the complicating issues I raised above) we can't say much after this age. However, if these animals might indeed live up to 103 years then 4400 days is only about 12% of the lifespan. I think it is unreasonable to conclude that this animal does not age when such a small portion of the lifespan has been studied.

We agree with the reviewer’s comment about it being expected for animals to survive longer in captivity than in the wild and provide considerable literature support of this principle. However even under the most ideal housing conditions for mice with ad lib food and in a pathogen free environment, mice show age-associated acceleration of mortality rates and to date none have lived to their allometrically predicted 6 year lifespan. But we disagree with the reviewer’s interpretation of our observations as covering an insufficient fraction of naked mole-rat lifespan to draw any conclusions. We previously tried to make the point that the concept of “maximum lifespan” is irrelevant to a species with constant hazard more akin to stochastic radioactive decay and presented the >100-year estimate as an illustration of that irrelevance. However, we were not very clear in stating that interpretation, so we have extensively edited and added to the Discussion to try and make our point clearer.

To summarize here: the time across which we observe no “aging” (i.e., no hazard increase) is far in excess of the typical lifespan of the organism in the wild. It is far beyond the expected full-lifespan for the species given its body-weight. And as we had previously discussed at length, it extends far, far beyond the age at which reproductive maturity can be attained (T_sex_). By any metric that could be used to estimate an expected lifespan for this organism, it not only exceeds that lifespan expectation, *but* exhibits no demographic sign of aging – i.e. no apparent approach to a lifespan limit – long after surpassing the expectation. We don’t think that the naked mole-rat’s exceptional longevity should appropriately be used as an argument against the mechanism by which it achieves that longevity.